# Missing wintertime methane emissions from New York City related to combustion

Luke D. Schiferl<sup>1</sup>, Andrew Hallward-Driemeier<sup>1,2</sup>, Yuwei Zhao<sup>1,2</sup>, Ricardo Toledo-Crow<sup>3</sup>, and Róisín Commane<sup>1,2</sup>

- 5 Lamont-Doherty Earth Observatory, Columbia University, Palisades, NY 10964, USA
  - <sup>2</sup>Department of Earth and Environmental Sciences, Columbia University, New York, NY 10027, USA
  - <sup>3</sup>Advanced Science Research Center, City University of New York, New York, NY 10031, USA

Correspondence to: Luke D. Schiferl (schiferl@ldeo.columbia.edu); Róisín Commane (r.commane@columbia.edu)

#### Abstract.

Accurately quantifying methane emissions from cities, and understanding the processes that drive these emissions, is important for reaching climate mitigation goals. Methane emissions from New York City metropolitan area (NYCMA), the most populous urban area of the United States, have consistently been underestimated by emission inventories compared to aircraft and satellite observations. In this study, we used continuous rooftop measurements of methane over 6 winter-to-spring transitions (January–May, 2019–2024) to examine the variability of city-scale methane enhancements (ΔCH<sub>4</sub>) and estimate methane emissions from the NYCMA. We found large variability in the 10-day mean observed ΔCH<sub>4</sub> (~50–250 ppbv) and monthly afternoon methane emissions rates (10.1–30.4 kg s<sup>-1</sup>) within and between the years of our study period. A recently released high-resolution regional methane emission inventory developed for the NYCMA performed better than other global and national inventories against the rooftop observations but still underestimated methane emissions, especially in winter. The estimates of methane emissions correlated with those of carbon monoxide (CO) emissions, determined from coincident measurements, suggesting a common city-scale incomplete combustion source for both methane and CO. Our analysis of these continuous measurements also implies a consistent diurnal cycle in urban methane emissions from the NYCMA, which reveals a potential bias in traditional afternoon-only approaches in this domain. This work highlights the usefulness of a long term, multi-species approach to constrain urban greenhouse gas emissions and their sources.

# 1 Introduction

Methane (CH<sub>4</sub>) is the second most potent greenhouse gas for climate change, with a global warming potential ~80 times greater than carbon dioxide (CO<sub>2</sub>) over 20 years (Forster et al., 2021). Atmospheric methane has a lifetime of only ~9 years (Prather et al., 2012) and thus provides a better opportunity than CO<sub>2</sub> for near-term mitigation of warming with emissions reductions (Jackson et al., 2020; Ocko et al., 2021; UNEP, 2021). The largest global anthropogenic sources of methane to the atmosphere are livestock production, the oil and gas industry, and landfills and other waste, while natural methane emissions come largely from wetlands (Saunois et al., 2020). However, the trends, magnitude, and variability of these methane emissions sectors

remain uncertain (e.g., Tibrewal et al. (2024); Turner et al. (2019)). Recently, methane emissions from oil and gas infrastructure (i.e., rural production facilities, pipeline leaks in cities) have received particular attention as mitigation targets (Alvarez et al., 2018; Ocko et al., 2021), highlighting the importance of accurately quantifying baseline methane emissions in order to track the effectiveness of mitigation efforts.

35

60

In cities, anthropogenic methane emission sources are expected to be limited to landfills, wastewater treatment plants, and natural gas distribution (pipelines), with some natural wetland emissions. Previous studies have identified atmospheric methane emissions that were greater than expected from inventories for cities across the world (e.g., in the United States of America (US): Boston (McKain et al., 2015; Sargent et al., 2021); Indianapolis (Lamb et al., 2016); Washington, DC (Ren et al., 2018); Los Angeles (Wunch et al., 2016); and Europe: Utrecht, Netherlands (Maazallahi et al., 2020); Hamburg, Germany (Forstmaier et al., 2023; Maazallahi et al., 2020); Munich, Germany (Chen et al., 2020); Bucharest, Romania (Fernandez et al., 2022)). Numerous studies from cities in the US have identified an unexpected correlation between methane emissions and natural gas consumption (e.g., He et al. (2019); Huang et al. (2019); Sargent et al. (2021)). Sargent et al. (2021) found methane emissions from Boston did not follow the distribution of natural gas infrastructure, and there was little decrease in emission rates over 8 years, despite concentrated efforts to mitigate leaks from pipelines. Without improved source attribution and understanding of urban methane emission processes, it is unlikely that cities will meet mitigation targets.

The New York City metropolitan area (NYCMA) is the densest and most populated urban region in the US and contains some of the oldest infrastructure of the country. Previous work to measure atmospheric methane in the NYCMA has used airborne data that have focused on snapshot time periods, particularly good weather days in the fall, winter, and spring. For example, using April and May 2018 aircraft observations, Plant et al., (2019) showed that the NYCMA was by far the largest urban source of methane across the northeast US and that the city emits 3–5 times more methane than estimated by the US national gridded inventory for 2012 (Maasakkers et al., 2016). Additional airborne measurements in November, February, and March over two winters (2018-2020) found methane emissions from the NYCMA to be 2.4 times higher than the same national inventory (Pitt et al., 2022). Analysis of flights from September 2017 and March 2018 indicated that the observed methane was more likely to be from natural gas than microbial sources around the NYCMA (Floerchinger et al., 2021).

The most recent US national gridded inventory (EPA GHGI v2023, Maasakkers et al., 2023) reduced the estimated methane emissions relative to the previous version (EPA GHGI v2016, Maasakkers et al., 2016) for the NYCMA, thus worsening the underestimate. EPA GHGI v2023 did not include natural gas post-meter methane emissions (assumed 100% combustion efficiency), but the simultaneously released EPA GHGI v2023 with Express Extension (EE) included a post-meter estimate that accounts for ~12% of total methane emissions in our study domain. A recently released higher-resolution regional inventory specific to the NYCMA indicated much greater methane emissions (~50% higher, including natural gas post-meter) than EPA GHGI v2023 but still underestimated airborne methane observations (Pitt et al., 2024b).

75

Satellite-based instruments have the potential to provide daily measurements of methane columns across large, diverse regions. However, these observations are limited to only clear sky afternoons, and current shortwave infrared instruments do not produce high-quality data over water, which presents a challenge for observing methane over coastal urban regions like the NYCMA, particularly surrounding the urban core of Manhattan Island. Still, Plant et al. (2022a) estimated the methane emissions from the NYCMA using TROPOMI data (methane and carbon monoxide (CO) column enhancement ratios) from 37 days of 2019 and found the mean emission rate to be 3-4 times larger than the EPA GHGI v2016, with a confidence interval spanning nearly twice the mean. The national-scale inversion performed by Nesser et al. (2024) using TROPOMI methane columns from 2019 found methane emissions for the NYCMA to be similar to the aircraft-constrained estimates from Pitt et al. (2022) and Pitt et al. (2024b). Continuous, in situ measurements bridge the gap in the observing system between airborne and satellite studies by providing additional temporal coverage through all weather and times of day.

In this study, we aimed to quantify and characterize the variability of city-scale methane enhancements ( $\Delta$ CH<sub>4</sub>) and emissions estimates from the NYCMA using continuous rooftop measurements from winter to spring over 6 years (2019-2024). Using an atmospheric transport model, we isolated the impacts of meteorology and emissions changes on the observed  $\Delta$ CH<sub>4</sub> and evaluated various global, national, and regional gridded methane emission inventories. We then identified changes to NYCMA methane emissions induced by the COVID-19 shutdown of spring 2020 and compared them with observation-informed estimates of coincident CO. Finally, we determined monthly methane and CO emissions estimates for our study period and domain and investigated the variability of these emissions over various timeframes to gain insight into the previously underestimated urban methane emissions sources.

#### 85 2 Methods

#### 2.1 In situ Observations

In this study, we used in situ observations of atmospheric methane abundance from a rooftop observatory in the dense urban core of the NYCMA and from a remote site located generally upwind of the city, which helped determine the abundance of methane entering the domain (i.e. the background).

# 90 2.1.1 Rooftop Measurements in the Urban Core

Ambient methane dry-mole fractions (units: ppbv, parts-per-billion by volume) were measured at the City University of New York Advanced Science Research Center (ASRC) Rooftop Observatory in Hamilton Heights, West Harlem, Manhattan (40.81534°N, 73.95033°W), a site located 56 m above ground level (93 m above sea level, a.s.l.) (Fig. S1). The CUNY ASRC site has been used extensively in recent years as a site representative of high-density urban air around the New York City metropolitan area. It has been the site of long-term studies (Schiferl et al., 2024), instrument characterization studies (Commane

et al., 2023, Khare et al., 2022), COVID activity change studies (Cao et al., 2023, Tzortziou et al., 2022) and was the location of atmospheric chemistry focused intensive studies in 2022 and 2023 (e.g. Hass-Mitchell et al., 2024 for the NYC-Mets project). The site sampled air most strongly interacting with the surface of a large area of Upper Manhattan and the Bronx and observed a mixture of methane from thermogenic and microbial sources including from natural gas infrastructure, wastewater treatment plants, and landfills. Additional details of the ASRC site were described in Commane et al. (2023) and Cao et al. (2023).




Several different instruments were used to measure dry-mole fractions of methane over the 6 consecutive winters and springs (January – May) of the study period (2019 – 2024) due to varying availability. The instruments used in this study were (i) Picarro G2401-m for 2019, 2020, and 1 January – 16 March 2023 (reporting at 0.5–1 Hz), (ii) Picarro G2401 for 2021, 16 March – 31 May 2023, and 2024 (reporting at ~0.3 Hz), and (iii) Aerodyne SuperDUAL for 2022 (reporting at 1 Hz). Each instrument was calibrated using gas cylinders that were traceable to standards calibrated by the Central Calibration Laboratory (CCL) at the National Oceanographic and Atmospheric Administration (NOAA) Global Monitoring Laboratory (GML) in Boulder, Colorado, USA. CCL maintains the World Meteorological Organization (WMO) methane scale (WMO CH4 X2004A). The Aerodyne SuperDUAL set-up at ASRC was described in Commane et al. (2023). Simultaneous measurements of dry-mole fractions of carbon monoxide (CO, calibration scale WMO CO X2014A) made at the ASRC site for 2019–2022 were described by Schiferl et al. (2024). Here we extended that record of CO measured at the ASRC site to include January–May 2023–2024.

We calculated the hourly mean methane dry-mole fraction at the ASRC site for hours with at least 50% valid sub-hourly observations (e.g., at least 1800 1-Hz measurements), which were rounded to the nearest 1 ppbv. Since we were interested in characterizing the methane variability of the entire NYCMA, rather than nearby sources, we removed the local-scale plume observations from the city-scale analysis. In the 1-Hz data, all examples of highly variable methane plumes (ie. near field sources) were strongly correlated with highly variable CO ( $R^2 > 0.99$ ). Methane observations were categorized as either city-120 or local-scale using the variability of the co-located CO observations at the ASRC site: hours with a CO standard deviation below 200 ppbv do not contain large plumes and were classified as city-scale. The threshold of 200 ppb for the CO standard deviation was chosen from a sensitivity analysis to replicate the results of the two-tower approach detailed in Schiferl et al., 2024. The categorization scheme indicated that many of the largest methane peaks were from local-scale sources near the observation site (Fig. S2), as was the case for CO in Schiferl et al. (2024). As these plumes are not representative of the broader 125 city scale, especially in 2020–2023, they were excluded from the analysis. The observed city-scale methane mole fractions had hourly peaks that were generally below 3000 ppbv and accounted for nearly 80% of the total observed hours. We also calculated the hourly mean CO at the ASRC site and classified hours of city-scale observed CO as for methane.

# 2.1.2 Remote Measurements to Constrain Domain Inflow

We used hourly methane dry-mole fractions for the entire study period from the Picarro G2301 on the Earth Networks tower in Stockholm, New Jersey (SNJ, 41.14356°N, –74.53872°W; 406 m a.s.l., 53 m above ground level intake height) as a paired remote background site (see Fig. S1, Sect. 2.2). The SNJ site was described by Karion et al. (2020), and the Earth Networks measurement system module was described by Welp et al. (2013) and Verhulst et al. (2017). All data were calibrated to the NOAA WMO calibration scale (WMO CH<sub>4</sub> X2004A), and data are archived at the National Institute of Standards and Technology (NIST) (Karion et al., 2025).

#### 135 **2.2 Observed Methane Enhancement Calculation**





We defined the observed methane enhancement (ΔCH<sub>4</sub>) from the NYCMA for each hourly city-scale observation as in Eq. 1:

observed 
$$\Delta CH_4$$
 = observed  $CH_4$  – background  $CH_4$  (1)

where the observed ΔCH<sub>4</sub> (units: ppbv) was the observed methane dry-mole fraction with the background methane removed. The background methane accounts for the atmospheric methane entering the study domain prior to being impacted by fluxes from the NYCMA.

To approximate the potential range in background methane, we estimated the rolling hourly 10-day background methane in two ways: (i) the fifth percentile of mole fractions at the urban core (ASRC) site using only the city-scale methane observations and (ii) the mean of the methane observations at the remote (SNJ) site, with both methods using data from the previous and following five days. These background estimation methods were applied as in Schiferl et al. (2024). We determined a confidence interval (CI) for each hourly background by calculating a distribution of backgrounds using a resampling bootstrap (n = 1000) with replacement over the methane observations for each rolling 10-day window. The background methane mole fractions were variable but most often peaked in late winter and declined toward June (Fig. S3). We also observed an increasing trend in background methane from year-to-year consistent with the increase in global atmospheric methane. The 95% CI for each hourly methane background was generally smaller, especially using the remote site method, than the variability in the background over time, which indicated high relative confidence in that background at a given hour. Differences in the background methane calculated from the two methods (an estimation of the background uncertainty) were up to 50 ppbv but were often much lower ( $\sim$ 5–10 ppbv). Given the position of the remote site in the prevailing upwind direction relative to the largest emitting regions of the NYCMA (Fig. S1), it is unlikely that the NYCMA was heavily sampled at the remote site, except for days with strong east winds. In this case, using the remote site as a background may lead to an underestimate in the magnitude of the observed  $\Delta$ CH4.

Observed  $\Delta CH_4$  was calculated for the ASRC site using the observed methane from that site and the distributions of both the urban core fifth-percentile background and the remote background. From the hourly observed ΔCH<sub>4</sub>, we calculated: (i) the 10day mean observed  $\Delta CH_4$  centered on each day of the study period, which allowed us to assess sub-monthly methane variability while removing variability on synoptic timescales, and (ii) the mean observed  $\Delta CH_4$  for each two-hour period throughout the day (a diurnal pattern) over various periods to assess sub-daily methane variability. These averaging techniques were 165 previously used by Schiferl et al. (2024) to assess the variability of CO from the NYCMA, but here we estimated the mean observed  $\Delta CH_4$  and corresponding CI by calculating a distribution using a resampling bootstrap (n = 1000) with replacement, where the sampled population included the distribution of backgrounds from both methods and the variability of observed methane within each averaging period. As in Schifferl et al. (2024), we only calculated the mean observed ΔCH<sub>4</sub> over averaging periods with at least 50% valid hours. We also recalculated and extended the record of observed ΔCO from Schiferl et al. 170 (2024) at the ASRC site using the urban core fifth-percentile and remote site mean backgrounds to match the time period of study and method for  $\Delta CH_4$  (now through 2024). For CO, the remote background was calculated using observations from the regional-scale Air Quality System (AQS) site operated by the Environmental Protection Agency (EPA) site at Cornwall, Connecticut (Fig. S1) as in Schiferl et al., (2024) since the methane remote site (SNJ) did not measure CO. The calibration of the EPA CO observations and their comparability to the ASRC observations are discussed in Schiferl et al., (2024). To avoid 175 biasing the corresponding distributions of mean observed  $\Delta CH_4$  and  $\Delta CO$ , we only used a given background site type (urban core or remote) in the distribution when both methane and CO data were available.

# 2.3 Methane Emission Inventories



We used anthropogenic methane emissions from 6 global, national, and regional inventories: 1) the global Emissions Database for Global Atmospheric Research (EDGAR) v6.0 for 2018 (Crippa et al., 2021), 2) the global EDGAR v8.0 for 2018 (Crippa et al., 2023, 2024), 3) the national EPA Greenhouse Gas Inventory (GHGI) v2016 for 2012 (Maasakkers et al., 2016), 4) US national EPA GHGI v2023 for 2018 (Maasakkers et al., 2023), 5) the national GHGI v2023 with Express Extension (EE) for 2018 (Maasakkers et al., 2023), and 6) the regional Pitt High-Resolution Inventory for 2019 (Pitt et al., 2024b). We used methane inventory emissions from the year 2018, which was the most commonly available year in our set of inventories, or from the closest year to 2018, when that year was not available. All methane emissions inventories used here were available monthly at 0.1°x0.1° spatial resolution, except for the Pitt High-Resolution Inventory, which presented an annual emissions rate at 0.02°x0.02° over a regional domain centered on the NYCMA. The Pitt High-Resolution Inventory used here was an ensemble comprised of 16 versions with varying scaling assumptions for the wastewater, stationary combustion, and natural gas distribution and post-meter sectors.

According to these inventories, landfills (24.3 – 52.8%), wastewater (11.9 – 29.4%), and natural gas distribution (8.6 – 26.1%) generally provided the largest annual sources of anthropogenic methane emissions from the NYCMA domain, while stationary combustion made up 5.7 – 10.9% of the domain total (Table 1). Spatially, landfill and wastewater emissions appeared as point

sources, while natural gas distribution emissions followed population density (Figs. S4, S5). Inventory emissions were greatest in the center of the NYCMA, in the densest urban infrastructure. The New York City (NYC) subdomain (Fig. S1) emitted ~30% of the NYCMA total methane emissions (Table S1). Large methane sources also existed away from the urban core as waste was transported for storage at suburban and rural landfill sites.

Table 1. Annual methane emissions from various inventories by sector and totals for the New York City Metropolitan Area (NYCMA) study domain. The NYCMA area is 46.7×10<sup>3</sup> km<sup>2</sup> and shown in Fig. S1. Methane emissions inventories are (left to right): Emissions Database for Global Atmospheric Research (EDGAR) v6.0 for 2018 (Crippa et al., 2021), EDGAR v8.0 for 2018 (Crippa et al., 2023, 2024), EPA Greenhouse Gas Inventory (GHGI) v2016 for 2012 (Maasakkers et al., 2016), EPA GHGI v2023 for 2018 (Maasakkers et al., 2023), GHGI v2023 with Express Extension (EE) for 2018 (Maasakkers et al., 2023), and Pitt High-Resolution Inventory for 2019 (Pitt et al., 2024b). For the Pitt High Resolution Inventory, emissions were the mean of the 16 ensemble versions.

| Methane Inventory Emissions | EDGAR  | EDGAR  | EPA GHGI | EPA GHGI | EPA GHGI | Pitt High-Res. |
|-----------------------------|--------|--------|----------|----------|----------|----------------|
| $[Gg CH_4 yr^{-1}]$         | v6.0   | v8.0   | v2016    | v2023    | v2023 EE | Inventory      |
| (Percentage of Total [%])   | 2018   | 2018   | 2012     | 2018     | 2018     | 2019           |
| Landfill                    | 126.0  | 123.5  | 102.7    | 66.7     | 67.2     | 64.6           |
|                             | (46.0) | (52.8) | (44.4)   | (37.2)   | (31.0)   | (24.3)         |
| Natural Gas Distribution    |        |        | 39.5     | 40.1     | 48.4     | 69.3           |
|                             | 23.6   | 23.2   | (17.1)   | (22.4)   | (22.3)   | (26.1)         |
| Natural Gas Transmission    | (8.6)  | (9.9)  | 14.5     | 15.1     | 16.8     | 10.2           |
|                             |        |        | (6.3)    | (8.4)    | (7.7)    | (3.8)          |
| Natural Gas Post-meter      |        |        |          |          | 26.1     | 52.3           |
|                             |        |        |          |          | (12.0)   | (19.7)         |
| Wastewater                  | 80.6   | 42.4   | 40.9     | 22.1     | 25.8     | 35.3           |
|                             | (29.4) | (18.1) | (17.7)   | (12.4)   | (11.9)   | (13.3)         |
| Stationary Combustion       | 15.7   | 25.6   | 15.3     | 17.3     | 17.2     | 15.4           |
|                             | (5.7)  | (10.9) | (6.6)    | (9.7)    | (7.9)    | (5.8)          |
| Other                       | 27.9   | 19.3   | 18.5     | 17.7     | 15.4     | 18.4           |
|                             | (10.2) | (8.3)  | (8.0)    | (9.9)    | (7.1)    | (6.9)          |
| Total                       | 273.8  | 234.1  | 231.4    | 179.1    | 216.9    | 265.6          |
| Total [kg s <sup>-1</sup> ] | 8.68   | 7.42   | 7.34     | 5.68     | 6.88     | 8.42           |

The total methane emissions and the relative contribution of source sectors varied greatly between the inventories. EPA GHGI v2023 had the smallest total methane emissions for NYCMA (5.7 kg s<sup>-1</sup>), while EDGAR v6.0 (8.7 kg s<sup>-1</sup>) had the largest total (Table 1). While the variability between the inventory totals was substantial (up to 3 kg s<sup>-1</sup>), this uncertainty was much smaller than the range of potential methane emission rates derived from previous observational studies (~10 kg s<sup>-1</sup>). EDGAR v6.0 had very high wastewater emissions compared to the other inventories, with twice the wastewater emissions from EDGAR v8.0 and four-times the wastewater emissions from EPA GHGI v2023. EDGAR v6.0 and v8.0 had larger landfill methane emissions than the other inventories, which were twice the landfill emissions from EPA GHGI v2023 and the Pitt High-Resolution Inventory. EDGAR v8.0 had ~50% higher stationary combustion methane emissions than the other inventories. EPA GHGI


v2023 EE and Pitt High-Resolution Inventory included methane emissions from post-meter natural gas, and the total emissions from that sector in both inventories were greater than the total natural gas distribution and transmission in EDGAR v6.0 and EDGAR v8.0, although the post-meter emissions in the Pitt High-Resolution Inventory were twice those in EPA GHGI v2023 EE. Generally, EDGAR v6.0 and EDGAR v8.0 had very small natural gas emissions components and larger relative landfill and wastewater emissions than the other inventories.




The differences in methane emissions between inventories were also evident in the spatial distribution of emissions throughout the domain. In the more densely populated NYC subdomain, EDGAR v8.0 had the smallest total methane emissions (1.4 kg s<sup>-1</sup>), while the Pitt High-Resolution Inventory had the largest (3.1 kg s<sup>-1</sup>) (Table S1). EDGAR v8.0 had more methane emissions from stationary combustion than from the wastewater, landfill, and natural gas sectors in NYC, while the natural gas component total alone from the Pitt High-Resolution Inventory was greater than the total for all sectors in EDGAR v8.0. EPA GHGI v2023 fell between EDGAR v8.0 and EPA GHGI v2016 in NYC total emissions and had more similar proportions by sector, but with lower wastewater and greater landfill emission totals, than the Pitt High-Resolution Inventory (and was missing postmeter natural gas completely). EPA GHGI v2023 EE (with post-meter natural gas) was more similar in totals and sector proportions to the Pitt High-Resolution Inventory but had half the post-meter emissions. The higher spatial resolution of the Pitt High-Resolution Inventory allowed for more precise positioning of emission sources within the NYC dense urban core. The spatial variability between some of the inventories may have been due to the incorrect gridding of point sources in some cases, such as the large point sources in New Jersey placed in adjacent grid boxes between inventories (Figs. S4, S5).

Monthly methane emissions changes in these inventories were minimal when applied over our January-May study period for the NYCMA. For example, EDGAR v6.0 and EPA GHGI v2023 varied less than 3% month-to-month compared to the mean annual rate. Monthly variability in EDGAR v6.0 was from stationary combustion emissions (5.7% of annual total), which dropped by more than 50% from January to May, while the monthly variability in EPA GHGH v2023 was from manure management (1.3% of annual total), which increased slightly only in May.

Compared to EDGAR v6.0 (Fig. S4), EDGAR v8.0 (Fig. S5) used updated spatial proxies for power generation, industrial facilities, and population distribution (Crippa et al., 2024). Scaling applied to these updated spatial proxies resulted in lower methane emissions in EDGAR v8.0 throughout the NYCMA and a different spatial distribution associated with population-dependent emissions such as wastewater and natural gas distribution. This change contrasts with the point source emissions from landfills which remained relatively constant between the two EDGAR versions.


EPA GHGI v2023 updated methane emissions totals for more recent years using methodological improvements and additional sources, and it better aligned gridding methods with underlying data sets than EPA GHGI v2016 (Maasakkers et al., 2023). In addition to including methane emissions from post-meter natural gas, EPA GHGI v2023 EE provided annual emissions

estimates consistent with US methane emission totals for each year but with the same spatial pattern proxy from EPA GHGI v2023 in 2018. Both EPA GHGI v2023 and EPA GHGI v2023 EE had less methane emissions from the NYCMA than EPA GHGI v2016, which previous studies have shown to be too low for this region (e.g., Plant et al. (2019)). Most of this methane reduction came from lower emissions from the landfill and wastewater sectors. For 2012, the only coincident year between the EPA GHGI versions, EPA GHGI v2023 was about 7% lower than EPA GHGI v2016 for the NYCMA and 27% lower for the NYC subdomain.



250

We did not apply any interannual emissions scaling to the inventories for our study period due to the large uncertainty of regional and city-scale variability, especially during the COVID-19 shutdown in 2020. Adding interannual variability to the inventories would have unnecessarily confounded the large differences that already existed between the inventories for the most common emissions year. While Crippa et al. (2020) suggested methods to implement diurnal variability in EDGAR using nationwide sector-specific scale factors, we did not apply a diel correction to the emissions of any inventory. Emissions for all inventories were constant throughout the day. Hourly methane emissions variability associated with stationary combustion was expected to be small. Methane emissions from natural sources (i.e., wetlands) are very limited during the winter and spring in the NYCMA, and we did not consider them here.

We also used monthly-varying CO emissions from EDGAR v8.1 (Crippa et al., 2024) for 2018, which were 15% higher on an annual basis for the NYCMA domain and 67% higher for the NYC subdomain than the EDGAR v6.1 CO emissions (Crippa et al., 2018, 2020) evaluated in Schiferl et al., (2024). EDGAR v8.1 included the same updated spatial proxies as in EDGAR v8.0 for methane (Crippa et al., 2024). We used CO emissions from EDGAR rather than from the EPA National Emissions Inventory (NEI) because at this time only EDGAR had both CO and CH<sub>4</sub> emissions, uniting the air quality and greenhouse gas emissions communities, as discussed in Schiferl et al. (2024). Hourly CO emissions variability from transportation combustion were expected to be much greater than that from stationary combustion, although we did not apply any hourly scaling to the CO emissions, consistent with our approach for methane.

# 2.4 Simulated Methane Enhancement Calculation

We simulated methane enhancements (ΔCH<sub>4</sub>) from the NYCMA for each hour of the study period as in Eq. 2:



simulated 
$$\Delta CH_4$$
 = inventory  $CH_4$  emissions flux × surface influence footprint (2)

where the simulated  $\Delta$ CH<sub>4</sub> (units: ppbv) was an inventory methane emissions flux (units: nmol m<sup>-2</sup> s<sup>-1</sup>) multiplied by the 24-hour surface influence footprint (units: ppbv (nmol m<sup>-2</sup> s<sup>-1</sup>)<sup>-1</sup>). The footprint is an indication of where and for how long the air interacted with the surface of the NYCMA in the previous 24 hours. We calculated simulated  $\Delta$ CH<sub>4</sub> using each of the 6 methane emissions inventories described in Sec. 2.3. We did not consider any loss of atmospheric methane over this 24-hour period

due to the long lifetime of methane (~9 years, Prather et al. (2012)), so all surface methane emissions intercepted by the footprint reach the observation site.

We calculated the surface influence footprint using the Stochastic Time-Inverted Lagrangian Transport (STILT) model driven by NOAA High-Resolution Rapid Refresh (HRRR) meteorology (3 km horizontal, hourly temporal resolution): together referred to as HRRR-STILT (Benjamin et al., 2016; Fasoli et al., 2018). STILT estimates the impact of surface gas fluxes on the atmospheric mole fraction by moving particles backward in time in three dimensions based on the HRRR winds and random turbulence. Interaction between the surface flux and atmospheric mole fraction (the surface influence) happens when particles are present within the lower half of the mixing layer. The accumulated surface influence of the particles was smoothed onto a regular 2-dimensional grid to form a surface influence footprint for ease of combination with the emissions flux inventories.





For this study, we derived the surface influence footprint at 0.01° horizontal and hourly temporal resolution for an integration period of 24-hours before the measurement at the ASRC observation site for each hour of the study period to match the hourly mean observations. Our configuration of HRRR-STILT for the NYCMA domain (Fig. S1) was previously used extensively to investigate CO and is described in more detail in Schiferl et al. (2024). While testing the configuration, Schiferl et al. (2024) found that the model configuration for vertical mixing and choice of meteorological product had little effect on the results at this site. They found that only the choice of the minimum Mixing Layer Height (MLH) produced a quantifiable change (> 1 ppbv) in the simulated CO mixing ratio; a 20 ppbv increase in simulated CO enhancement was observed when reducing the minimum MLH from 250 m to 150 m (Figure S7 in Schiferl et al., 2024). We evaluated four possible parameterization of the MLH in STILT and all configurations simulated methane enhancements that differed by less than 1 ppbv in the afternoon, increasing to a maximum of 5 ppbv at night. We also tested the impact of the STILT minimum mixing height (150m v. 250m) and meteorological product (HRRR v. NAMS, North American Mesoscale Forecast System at 12 km horizontal resolution) on our monthly observation-informed emissions estimates (see Sec. 2.5) for 2023 and 2024 and discuss those sensitivity results in Sec. 3.3.

The surface influence footprint from each hourly HRRR-STILT simulation combined with the inventory methane emission flux produced a single simulated  $\Delta$ CH<sub>4</sub>, which we matched with the valid hourly observed  $\Delta$ CH<sub>4</sub> at the ASRC site. Mean simulated  $\Delta$ CH<sub>4</sub> and a corresponding distribution was calculated from the hourly simulated  $\Delta$ CH<sub>4</sub> as described above for the mean observed  $\Delta$ CH<sub>4</sub> (over 10-day and 2-hour periods). For the Pitt High-Resolution Inventory, the distribution of simulated  $\Delta$ CH<sub>4</sub> included the ensemble of 16 inventory versions. We also calculated hourly and mean simulated  $\Delta$ CO using the same HRRR-STILT footprints and CO emissions from EDGAR v8.1.

# 2.5 Observation-informed Methane Emissions

We calculated observation-informed methane emissions estimates from the NYCMA for each month (or various multi-week periods during the COVID-19 shutdown, see Sec. 3.3) of the study period as in Eq. 3:

observation-informed CH<sub>4</sub> emissions flux = domain total inventory CH<sub>4</sub> emissions flux 
$$\times \frac{\text{observed }\Delta\text{CH}_4}{\text{simulated }\Delta\text{CH}_4}$$
 (3)

where the distribution of methane emissions was determined using valid hourly observed  $\Delta$ CH<sub>4</sub> and simulated  $\Delta$ CH<sub>4</sub> (using annual emissions from the Pitt High-Resolution Inventory) sampled from afternoon (11–16 h EST) hours only and from all hours (24-hr). Afternoon emissions estimates required at least 30 valid observation hours, and 24-hour estimates required 144 valid observation hours per multi-week period (minimum 6 observations per hour length) to be calculated. This calculation used the relative bias in the methane inventory compared to the methane observations to adjust the initial emissions inventory, and when applied over multi-week timescales to widely sample the study domain, estimated a city-scale methane emissions flux for the NYCMA. A similar method was used to calculate afternoon methane emissions for Boston, Massachusetts by Sargent et al. (2021). We combined the retained distributions of the previously calculated hourly observed and simulated  $\Delta$ CH<sub>4</sub> such that the resulting observation-informed methane emissions and corresponding CI (calculated at 50% and 95%) account for the background uncertainty, the variability of the observed methane mole fractions within each period, and the ensemble estimate from the Pitt High-Resolution Inventory.




We also calculated the observation-informed CO emissions and corresponding CIs using hourly observed  $\Delta$ CO and simulated  $\Delta$ CO (using emissions from EDGAR v8.1) using the same method as for methane. As with calculating the mean 10-day and 2-hour observed and simulated enhancements, we only used a given background site type (urban core or remote) in the distribution when the background site type was valid for both methane and CO.


Aggregating hours over the afternoon hours, when the atmospheric transport and mixing is less uncertain, and over the entire day, to increase observational coverage in time and space, provided more confident estimates compared to shorter or more uncertain time periods (e.g., 2-hour periods, overnight). These longer aggregation time periods resulted in much narrower confidence intervals, boosting the confidence in our observation-informed emission rates.

#### 340 2.6 Carbon Monoxide (CO) as a Combustion Tracer

We used coincident observations of CO from the ASRC site as a tracer for incomplete combustion. CO is emitted as a byproduct of combustion when the efficiency of burning a carbon-based fuel source is not optimized, with higher CO emissions per amount of fuel burned indicating a more inefficient combustion process. In the US, CO emission rates have been declining due to improvements in on-road vehicle efficiency, the largest source of CO emissions nationwide (e.g., Hedelius et al., 2021;

Lopez-Coto et al., 2022; Yin et al., 2015). Generally, CO emission sources are not co-located with large urban methane emission sources such as landfills, wastewater treatment, and natural gas distribution as the methane from these sources is not actively being combusted, and methane and CO sources are not linked in emissions inventories (outside of wildfires and wood burning). In a well-controlled environment, methane can be efficiently burned, but inefficient combustion can lead to large methane emissions (Plant et al., 2022b). Post-meter methane emissions are thought to be from leaks in the local system and do not have well-documented corresponding CO emissions.

Schiferl et al. (2024) characterized CO emissions from the NYCMA using a shorter period of observations from the ASRC site (ending in 2022). That study found large variability in city-scale observed  $\Delta$ CO,  $\sim$ 60% of which was driven by atmospheric transport meteorology and  $\sim$ 40% of which was driven by emissions changes. Schiferl et al. (2024) also found a substantial underestimate in simulated  $\Delta$ CO when evaluating CO inventory emissions from EDGAR v6.1 and that the observed  $\Delta$ CO and associated CO emissions from the transportation sector were unlikely to account for the observed  $\Delta$ CO variability and magnitude outside of the COVID-19 shutdown of spring 2020.

In this study, we extended the record of hourly observed CO dry-mole fractions at the ASRC site to match the record of methane observations described above. We excluded hours identified as local-scale observations, removed urban site and remote backgrounds, and calculated simulated ΔCO and observation-informed CO emissions in the same manner as for methane. We used the EDGAR v8.1 CO inventory emissions combined with the observed-to-simulated ΔCO ratio to estimate the city-scale CO emissions. Since CO emissions in EDGAR v8.1 were greater than in EDGAR v6.1, especially in the urban core, simulated ΔCO driven by EDGAR v8.1 were improved over the EDGAR v6.1 CO emissions evaluated by Schiferl et al. (2024) when compared to the observed ΔCO from the ASRC site.

#### 3 Results and Discussion



We first use our rooftop observations from six years of winter-to-spring transitions to quantify the magnitude and variability of the city-scale observed methane enhancements ( $\Delta$ CH<sub>4</sub>) from the New York City metropolitan area (NYCMA) and their correlation with enhancements from incomplete combustion ( $\Delta$ CO). Then, we use our simulations to evaluate and identify bias in various regional-to-global scale methane emission inventories and remove variability in the observations from atmospheric transport (meteorology). Next, we examine diurnal variability in the observed and simulated  $\Delta$ CH<sub>4</sub> and quantify the changes in methane emissions that occurred relative to known CO emissions declines during the COVID-19 shutdown of spring 2020. Finally, we present monthly observation-informed methane and CO emission rate estimates for the NYCMA over the study period and discuss potential reasons for their correlation.

# 375 3.1 City-scale Observed ΔCH<sub>4</sub>



The observed  $\Delta$ CH<sub>4</sub> from the NYCMA varied substantially on sub-monthly timescales throughout the winter and spring across all years of the study (Fig. 1). Mean 10-day observed  $\Delta$ CH<sub>4</sub> ranged from ~50 ppbv to ~250 ppbv. The winters of 2019, 2020, 2022, and 2023 experienced extended large peaks (>100 ppbv) in observed  $\Delta$ CH<sub>4</sub> with general declines toward spring. The large peak in 2021 occurred in late March, at the beginning of the transition to spring, while several moderate peaks (50–100 ppb) were observed in winter 2024. In 2019, 2020, and 2023, there was less variability outside of these extended peaks as compared to 2021, 2022, and 2024, which showed several additional small episodes of more elevated  $\Delta$ CH<sub>4</sub> (~50 ppbv).

Observed  $\Delta$ CO from the NYCMA also varied substantially throughout the study period (Fig. 1), as previously shown by Schifferl et al. (2024) for 2019-2022. The 10-day mean observed  $\Delta$ CH<sub>4</sub> and observed  $\Delta$ CO varied together throughout the study period (Fig 1), except for during the COVID-19 shutdown of 2020 (see Sec 3.3). There was a strong correlation between the observed enhancements of both species (Fig. 2a, R<sup>2</sup> = 0.61), with generally higher observed  $\Delta$ CH<sub>4</sub> and observed  $\Delta$ CO during winter (January–February) than in spring (April–May). A large portion of the correlation is likely from the variability of atmospheric transport but could also indicate simultaneous emission sources of both methane and CO.

Figure 1. Timeseries of 10-day mean observed  $\Delta CH_4$  (black) and  $\Delta CO$  (red) for the New York City Metropolitan Area (NYCMA) domain at the urban core ASRC site during January–May 2019–2024. Vertical bars show the 95% confidence interval (CI) of the mean. Observations are plotted in time at the center of the 10-day averaging period. The COVID-19 shutdown period (15 March–31 May 2020) is shaded in blue.

The uncertainty in the 10-day mean observed  $\Delta$ CH<sub>4</sub> derived from the different background methane calculation methods was most often ~10–25 ppbv but spanned near 0 ppbv to 50 ppbv. This uncertainty varied between time periods. For example, the uncertainty in observed  $\Delta$ CH<sub>4</sub> was notably small throughout 2019 and 2020, while larger uncertainty occurred during March 2022 and more consistently throughout 2023. When combining the uncertainty in the background with the variability of observed methane mole fractions within each averaging window, the 95% CI of the 10-day mean consistently spanned 20-50 ppbv, with some CI reaching nearly 100 ppbv. The 95% CI of the 10-day mean observed  $\Delta$ CO were usually similar to, or slightly smaller than, those for observed  $\Delta$ CH<sub>4</sub>. The 95% CI for both species were also mostly smaller than the variability over time in the 10-day mean, which indicates confidence that we can detect changes in observed  $\Delta$ CH<sub>4</sub> and  $\Delta$ CO on the 10-day timescale.

Figure 2. (a) Comparison of 10-day mean observed ΔCH<sub>4</sub> and ΔCO at the urban core ASRC site as in Fig. 1 for the study period colored by day of year. Horizontal (ΔCH<sub>4</sub>) and vertical (ΔCO) bars show the 95% CI of the mean. The linear best fit line, slope, and uncertainty from standard error determined by York fit, the coefficient of determination (R<sup>2</sup>), and the number of points considered (N) are shown as indicated. The 1:1 line is shown in dark gray. (b) Comparison of 10-day mean observed ΔCH<sub>4</sub> and ΔCO as in (a) separated and colored by COVID-19 shutdown (15 March–31 May 2020; red) and non-shutdown (all other times; black) periods.

#### 3.2 Evaluation of NYCMA Methane inventories




We found much more variability in observed ΔCH<sub>4</sub> for the NYCMA than could be explained by existing emissions inventories. Monthly methane emissions for the NYCMA from EDGAR v8.0 and EPA GHGI v2023, the most recent global and US national inventories, only declined by 0.5% and 2.7%, respectively, between their seasonal maximum and minimum. For the

smaller sub-domain over NYC, the two inventories declined by a similarly small rate (0.9% for EDGAR v8.0, 1.2% for EPA GHGI v2023). The Pitt High-Resolution Inventory had no sub-annual variability. Emissions inventories are designed to be longer term snapshots of average emissions and cannot accurately account for all mechanistic variations in emissions processes.

We compared the 10-day mean observed ΔCH<sub>4</sub> with the corresponding simulated ΔCH<sub>4</sub> to evaluate the magnitude in the inventories and to partition the sources of the observed ΔCH<sub>4</sub> variability between meteorology and emissions changes. All six methane emission inventories we examined consistently underestimated the observed ΔCH<sub>4</sub> from the NYCMA (Figs. 3, S6), and the degree of performance generally followed the domain-wide totals for each inventory (larger emissions performed better).

**Figure 3.** Timeseries of 10-day mean observed ΔCH<sub>4</sub> (black) for the NYCMA domain as in Fig. 1 and simulated ΔCH<sub>4</sub> determined by HRRR-STILT combined with methane emissions from EDGAR v6.0 (blue), EPA GHGI v2023 (brown), and Pitt High-Resolution Inventory (purple). Vertical bars show the 95% CI of the mean.

The Pitt High-Resolution Inventory (Fig. 4a) performed the best of the inventories evaluated, having the smallest underestimate in simulated  $\Delta$ CH<sub>4</sub> (slope = 0.60±0.05) for the entire study period (Fig. 4b). The comparison differed seasonally, with most of the missing observed  $\Delta$ CH<sub>4</sub> occurring during with winter, in contrast with the inventory matching or even overestimating the observed  $\Delta$ CH<sub>4</sub> in the spring (Fig. 4b). The Pitt High-Resolution Inventory coupled with HRRR-STILT also captured peaks and variability in the observed  $\Delta$ CH<sub>4</sub> not captured by other models (such as in March 2019 and April 2023) (Fig. 3). Simulated  $\Delta$ CH<sub>4</sub> using EDGAR v6.0 had a slightly greater underestimate (slope = 0.51±0.04) compared to the observed  $\Delta$ CH<sub>4</sub> despite a slightly higher domain-wide methane emissions total than the Pitt High-Resolution Inventory. However, the Pitt High-Resolution Inventory emissions for the NYC sub-domain were 32% higher than for EDGAR v6.0 in this region, the area of the domain most heavily sampled by atmospheric observations (Fig. 4a). These discrepancies highlight the importance of accurate and highly resolved spatial emissions distributions for a city with highly variable and heterogeneous sources (Tables 1 and S1).

**Figure 4. (a)** Map of the annual mean methane (CH<sub>4</sub>) emissions flux from the Pitt High-Resolution Inventory (Pitt et al., 2024b) for the NYCMA study domain, the locations of the urban core ASRC and remote SNJ observations sites used in the study, and contours of the  $50^{th}$  (solid) and  $75^{th}$  (dashed) percentile mean surface influence footprint from HRRR-STILT used to calculate the mean 10-day simulated ΔCH<sub>4</sub> in **(b)** for the entire study period. **(b)** Comparison of 10-day mean observed and simulated ΔCH<sub>4</sub> at the ASRC site as in Fig. 3 colored by day of year, where simulated ΔCH<sub>4</sub> was calculated using the Pitt High-Resolution Inventory. Observed ΔCH<sub>4</sub> are plotted as in Fig. 2. Simulated ΔCH<sub>4</sub> are plotted with horizontal bars for the 95% CI of the mean. Statistics and annotation are as in Fig. 2.

The EDGAR v6.0 methane inventory performed the best of the global and national inventories compared to the atmospheric observations and the simulated  $\Delta$ CH<sub>4</sub> were considerably better in magnitude than the more recent inventories such as EDGAR v8.0 (slope = 0.35±0.03) and EPA GHGI v2023 (slope = 0.29±0.02). The previous US national inventory, EPA GHGI v2016 (slope = 0.36±0.03), had a smaller underestimate than the newer version, however, including post-meter emissions in EPA GHGI v2023 EE (slope = 0.39±0.03) improved the performance of the updated EPA inventory considerably.

Our atmospheric observations thoroughly sampled all directions throughout the domain for the study period, according to the surface influence footprints from our transport model simulations, with a slight preference to the southern half of the domain (Fig. 4a). Accounting for the varying atmospheric transport and vertical mixing throughout the study period, which drives nearly all variability in the simulated  $\Delta$ CH<sub>4</sub>, we found that atmospheric transport and vertical mixing only explained 30%–43% of the variability in observed  $\Delta$ CH<sub>4</sub>, depending on inventory comparison, based on the calculated R<sup>2</sup> between the observed  $\Delta$ CH<sub>4</sub> and simulated  $\Delta$ CH<sub>4</sub>. We note that the Pitt High-Resolution Inventory with no monthly emissions variability was in the middle of this range (R<sup>2</sup> = 0.34), indicating that incorporating the monthly emission changes included in the other inventories had limited impact on the outcome. We found that the impact of atmospheric transport and vertical mixing on observed  $\Delta$ CH<sub>4</sub> in this study was considerably less than was found by Schiferl et al. (2024) for CO using the same metric and largely same methods, where ~60% of the variability in observed  $\Delta$ CO was due to atmospheric transport and vertical mixing. The weaker correlation for methane than CO implies that the methane emissions may change more across the seasons when calculated on a 10-day time scale. This result is consistent with relatively unchanging seasonal magnitudes of CO emissions from traffic,

power generation, and manufacturing, which are sources that are not expected to contribute much to the methane emissions totals in the NYCMA.

# 3.3 NYCMA Methane Emissions during the COVID-19 Shutdown




We found that the COVID-19 shutdown of spring 2020 had limited impact on the observed ΔCH<sub>4</sub> from the NYCMA (Fig. 2b). The 10-day mean observed ΔCH<sub>4</sub> centered on 15 March–31 May 2020 were not outside the distribution of observed ΔCH<sub>4</sub> for other study time periods. This result contrasts with the observed ΔCO for the same shutdown period, which decreased by up to 50 ppb below the lower end of the distribution for non-shutdown periods (Fig. 2b). The COVID-19 shutdown was coincident with meteorological conditions favoring lower surface influence (as described by Schiferl et al. (2024)), and so the observed enhancements of both species were on the lower end of the observations for the entire study period, but there was no clear step change decrease in observed ΔCH<sub>4</sub> like there was for observed ΔCO.

Using our continuous hourly data record, we also examined the changes in the diurnal pattern of  $\Delta CH_4$  prior to and during the COVID-19 shutdown (Fig. 5a). The mean diel cycle of observed  $\Delta CH_4$  and simulated  $\Delta CH_4$  for the NYCMA both generally followed the height of the mixing layer:  $\Delta CH_4$  peaked in the early morning hours when the layer was lowest, decreased throughout the day as the layer rose, and increased again in the evening. The simulated  $\Delta CH_4$  using the Pitt High-Resolution Inventory were generally lower than the observed  $\Delta CH_4$ , especially during the daytime hours in winter periods prior to the COVID-19 shutdown, consistent with the mean 10-day underestimate identified above.

Figure 5. (a) Diurnal timeseries of mean observed  $\Delta$ CH<sub>4</sub> (black) and simulated  $\Delta$ CH<sub>4</sub> (purple) using Pitt High-Resolution Inventory for the NYCMA domain at the urban core ASRC site.  $\Delta$ CH<sub>4</sub> were averaged every two hours for various periods before and during the peak COVID-19 shutdown (15 January–15 May 2020, left to right). Vertical boxes show the 50% CI and vertical bars show the 95% CI of the mean. (b) Diurnal time series (black) of the ratio of observed  $\Delta$ CH<sub>4</sub> to simulated  $\Delta$ CH<sub>4</sub> from (a) and the ratio of mean afternoon (11–16h) and mean 24-hr observed to simulated  $\Delta$ CH<sub>4</sub> (blue/black) and  $\Delta$ CO (red) for each period and plotted in the same matter as in (a). Simulated  $\Delta$ CO was calculated using CO emissions from EDGAR v8.1.

Since there was no diurnal variability in the inventory methane emissions, the diurnal variability in the simulated  $\Delta$ CH<sub>4</sub> was entirely due to changes in the surface influence footprint (i.e., atmospheric transport and vertical mixing) throughout the day. Sensitivity studies of the simulated footprints found that simulated methane could change by up to 1ppb during the afternoon to 5 ppb overnight depending on the configuration of the model. However, the diel changes in the methane enhancements were of the order 80-200 ppb at night (Figure 5a) so we estimated the model bias was at most 6%. The differences in the variability between the observed  $\Delta$ CH<sub>4</sub> and simulated  $\Delta$ CH<sub>4</sub> were therefore attributed to changes in the methane emissions which were not included in the inventory. By normalizing the observed  $\Delta$ CH<sub>4</sub> by the simulated  $\Delta$ CH<sub>4</sub>, we minimized the impact of meteorology, thereby isolating only the changes in methane emissions. Using this method, Schifferl et al. (2024) found that most of the CO emissions changes occurred in areas located within 2 hours atmospheric transport of the ASRC site and we expect a similar atmospheric transport time for methane. This normalization method produced observation-informed changes in methane emissions for multi-week periods before and during the COVID-19 shutdown (Fig. 5b).






Prior to the COVID-19 shutdown, we found that the normalized ΔCH<sub>4</sub> exhibited a large diurnal cycle with a peak at midday and consistent minimum overnight (Fig. 5b). The daytime peak degraded slightly for the early part of the shutdown (15–31 March), nearly disappeared for early April and returned for the last month (15 April–15 May) of the shutdown. These observed pattern changes imply methane emissions variability that occurred throughout the day and emissions changes that occurred during different time periods of the COVID-19 shutdown. Although not examined closely, these daytime diurnal peaks in normalized ΔCH<sub>4</sub> consistently occurred for all months of the study period.

As part of estimating the observation-informed methane emissions rates when combined with the Pitt High-Resolution Inventory (see Sec 2.5), we calculated the aggregated 5-hour afternoon (11–16 h EST) and 24-hour daylong ratios of observed-to-simulated  $\Delta$ CH<sub>4</sub>. These aggregated normalized  $\Delta$ CH<sub>4</sub> were similar to the ratios of the coincident 2-hour time periods but were produced with much narrower confidence intervals (Fig. 5b). The observed-to-simulated  $\Delta$ CO ratio for the same time periods showed a larger relative decrease in afternoon when compared to normalized  $\Delta$ CH<sub>4</sub> (Fig. 5b), consistent with the expected larger decrease in CO emissions (likely from the transportation sector) due to the COVID-19 shutdown.

Afternoon observation-informed methane emission rates from the NYCMA decreased by 22% (16.2 to 12.6 kg s<sup>-1</sup>) between early March and the COVID-19 shutdown of late March 2020 (Table S2). Afternoon CO emission rate reductions were much greater, 49% (44.9 to 22.9 kg s<sup>-1</sup>) between the same time periods. Clearly, the large reduction in CO emissions was at least partly due to large reductions in the transportation sector due to stay-at-home orders, as expected and observed in other cities (Lopez-Coto et al., 2022; Monteiro et al., 2022). However, Schiferl et al. (2024) also showed that, for the NYCMA, the observed reduction in traffic was not enough to fully explain the reduction in observed ΔCO due to COVID-19 shutdowns. Therefore, it is possible that the methane and CO emissions reductions during the COVID-19 shutdown, from sources other than transportation, were related. It remains uncertain if these COVID-19 shutdown reductions were due to an activity change

(e.g., urban population decline, work from home policies) or merely corresponded to seasonal or other non-shutdown emissions mechanisms over the COVID-19 shutdown (e.g., reduction in building heating due to warmer weather).

# 3.4 Monthly NYCMA Observation-informed Methane Emission Rates


Meteorological products (i.e., 3-dimensional wind fields) used to drive the atmospheric transport model are more uncertain at night, for mixing heights especially, and so we focused first on monthly emissions estimates using only afternoon hours. We found the monthly afternoon observation-informed methane emission rates for the NYCMA were highly variable over our study period (Fig. 6a). Generally, the methane emission rates had large, variable peaks in the winter, plateaued during the winter-to-spring transition, and fell to seasonal lows by May. The 95% CI of these methane emission rate estimates, which included background uncertainties, variability in the observations, and an ensemble of inventory configurations, also varied widely, spanning a range of 4 to 17 kg s<sup>-1</sup>. The greatest methane emission rate occurred in January 2021 (30.4 kg s<sup>-1</sup>), with the lowest methane emission rate in May 2022 (10.1 kg s<sup>-1</sup>; May 2023 and May 2024 are very similar), excluding the COVID-19 shutdown of 2020. Most of the afternoon observation-informed methane emissions rates were much larger than the best-performing emissions inventory, the Pitt High-Resolution Inventory (8.4 kg s<sup>-1</sup>). The surface influence footprints used in these afternoon estimates generally sampled the NYCMA domain consistently for all months of the study period (Fig. S7).

The NYCMA observation-informed CO emission rates for the afternoon over the same time periods (Fig. 6b) showed similar trends in variability to those of methane, but without the extreme January peaks. The CO emissions rates for January 2021 and February 2019, for example, were also times of large methane emission rates. However, several March and April CO emission rates were high, while methane was reduced relative to the cold months. This difference could be due to the relatively large portion of CO emissions from non-heating related sources that are expected to be consistent throughout the winter-to-spring transition (e.g., transportation).

The impact of the COVID-19 shutdown on atmospheric composition was clearly seen in the monthly estimates as well (Figs. 6a–b). A nearly linear month-to-month decrease in emission rates between February and May 2020 resulted in observation-informed CO emissions reductions of 73%, which was only slightly larger than the relative reduction in methane emissions (67% over the same period). Afternoon observation-informed emissions rates for the NYCMA for methane and CO are in Tables S3 and S4, respectively.

**Figure 6.** Monthly afternoon (11–16 h) observation-informed (a) methane (CH<sub>4</sub>) and (b) CO emission rates from the NYCMA for January–May 2019–2024. Vertical boxes show the 50% CI and vertical bars show the 95% CI of the emission rate. Monthly emission rates are colored by year according to the legend. Emission rates from 2020 are outlined in black. Individual months without enough data to meet availability threshold to calculate a monthly emission rate are not shown. (c) Comparison of monthly afternoon observation-informed methane and CO emission rates as in (a–b). Horizontal (CH<sub>4</sub>) and vertical (CO) bars show the 50% CI of the emission rate. Monthly emission rates are colored by month according to the legend. The R<sup>2</sup> is shown as indicated. (d) Observation-informed methane emission rates from this study (afternoon rooftop observations) compared to methane emission rates for the NYCMA from other studies using aircraft and satellite observations. For this study, the horizontal box shows the 50% CI and the horizontal bar shows the 95% CI of the mean emission rate. For other studies, the definition of emission rate point and horizontal bars varies by study-specific method.

The afternoon observation-informed methane and CO emissions rates for the NYCMA were well correlated over our study period (Fig. 6c, R<sup>2</sup> = 0.59). Unlike the observed ΔCH<sub>4</sub>:ΔCO dry mole fraction enhancements compared in Sec. 3.1, this relationship between methane and CO emissions accounted for variability in atmospheric transport. We do not know the CH<sub>4</sub>:CO emission ratio, nor the modified combustion efficiency of individual incomplete combustion sources (i.e. boilers and other appliances) within our study domain. However, we can expect the CH<sub>4</sub>:CO ratio to be variable with each appliance configuration, and so it may change across time and space. We likely observed two competing thermogenic methane source sectors at the rooftop: (1) inefficient consumption of natural gas during peak heating season (January-February), which is correlated with extreme cold events, and (2) intermittent emissions of natural gas during the appliance duty cycle (also known as "slip") (Lindburg et al., 2025). During the winter-to-spring transition, when outdoor temperatures vary around 55°F, the threshold below which all buildings are required to be heated by NYC laws (Chapter 2: Housing Maintenance Code, 2025),

boilers will repeatedly cycle. This evidence suggests a common source of methane and CO emissions, which may be related to stationary incomplete combustion. Further study is required to isolate and quantify these processes in more detail.






Our estimates for NYCMA afternoon methane emission rates overlap with methane emissions estimates from previous airborne and satellite studies, which also focused on the afternoon period only (Fig. 6d). Our long-term in situ measurements spanned a greater range of methane emissions, especially at the high end of studies using in situ measurements. Regional inversions using aircraft data by Pitt et al. (2022) and Pitt et al. (2024b) and a national inversion using satellite data by Nesser et al. (2024) found optimized methane emissions from the NYCMA on the lower end (9.4–10.5 kg s<sup>-1</sup>) of our estimates but with narrow uncertainty. Plant et al. (2019) used CH<sub>4</sub>:CO<sub>2</sub> and CH<sub>4</sub>:CO<sub>2</sub> ratios from aircraft data and inventories to estimate methane emissions for the NYCMA and found similar mean emissions estimates using both methods close to our mean estimate, but their estimate using CH<sub>4</sub>:CO had a much larger uncertainty range, which is similar to our range that combines uncertainty and variability throughout our study period. Plant et al. (2022a) used satellite column CH<sub>4</sub>:CO ratios to estimate methane emissions, the uncertainty of which spanned our entire range of emissions estimates. All these airborne and satellite studies used the US Census Bureau Topologically Integrated Geographic Encoding and Referencing (TIGER) domain for New York–Newark, which contains ~70% of the total emissions of Pitt High-Resolution Inventory used in our study. We note that the airborne studies were restricted to weather conditions that are suitable for flight and satellite studies were restricted to clear-sky days when the methane plume did not move out to sea.

We tested the sensitivity of our observation-informed emissions estimates to assumptions in the transport model in two ways:

1) by lowering the minimum mixing height from 250m (default) to 150m when STILT is driven by HRRR, and 2) by driving STILT with NAMS instead of HRRR. We found a consistent reduction in afternoon emissions estimates when lowering the HRRR minimum mixing height, with a mean decrease of 13% for methane and 10% for CO emissions over 2023 and 2024 (Fig. S8). Using NAMS meteorology resulted in similar drops in mean methane (13%) and CO (16%) emissions estimates, but with a larger range of changes across months (~50% reduction to ~25% increase in individual months) which weakened the winter-to-spring emissions relationship compared to the default HRRR-STILT configuration. Given the heterogeneity of the complex NYCMA landscape, the differences in emissions estimates are more likely to be due to the spatial resolution of meteorological product (HRRR: 3km, NAMS: 12km) than mixing height errors for the afternoon time periods.

The methane and CO emission estimates for this study (Fig. 6) and in previous studies used afternoon observations only, because there is greater confidence in the atmospheric transport processes used to interpret the observations during this time of day. When we calculated the monthly observation-informed methane and CO emission rates using all observations for a given month (24-hour rate), we found consistently lower emissions rates for both methane and CO (Figs S9a-b). The 24-hour emission rates for methane and CO were similarly correlated (R<sup>2</sup> = 0.53) as they were using afternoon hours only and maintained a similar winter-to-spring decline (Fig. S9c). The consistent difference between the afternoon and 24-hour emission

rates suggests a diurnal cycle in emissions, which is well-known for CO emissions (traffic, human activity), but had not been, to our knowledge, previously inferred for urban methane emissions. These diurnal emissions patterns were not included in methane inventories nor in the CO inventories used here. If related to combustion, the methane and CO emissions from building heating sources could be greater during the day when commercial and industrial buildings increase heating temperature as occupancy increases. The combination of many hours to produce the monthly 24-hour emissions estimates resulted in narrower 620 confidence intervals compared to the afternoon-only emission estimates, further supporting the possibility of diurnal cycle for urban methane emissions. The 24-hour observation-informed methane emissions estimates were also more consistent with the inventories evaluated here. CO emissions studies (e.g., Lopez-Coto et al. (2022)) apply a daytime correction given the assumed diurnal pattern of CO emissions, so a similar correction may be needed for methane as well to avoid biasing observationalconstrained methane emissions too high. The surface influence footprints used in the 24-hour emission rate estimates were 625 more balanced in all directions and more contained within the NYC subdomain than when using only the afternoon hours (Figs. S7, S10), and this implies more sensitivity to larger emissions sources in the urban core on average. The 24-hour observation-informed emissions rates for the NYCMA for methane and CO for each month January – May for 2019 – 2024 are shown in Tables S5 and S6, respectively. Our sensitivity analysis of the atmospheric transport model for the 24-hour emissions estimates found that reducing the minimum mixing height consistently lowered the estimated emissions for the 630 NYCMA (mean CH<sub>4</sub> by 21%, mean CO by 19%) (Fig. S8). Using NAMS reduced the mean 24-hour estimated emissions by similar relative amounts (CH<sub>4</sub> by 15%, CO by 20%), although the impact of NAMS on emissions ranged from ~30% reduction to ~15% increase depending on the month across both species.

# 4 Conclusions

645

Using in-situ rooftop observations, this study found unexpected variability in atmospheric methane mole fractions, city-scale enhancements, and methane emission rates from the New York City Metropolitan Area (NYCMA) over 6 winter-to-spring transition periods. Our work reveals the power of long-term continuous measurements, since this variability is not captured by favorable weather-only aircraft campaigns or afternoon, clear-sky satellite measurements. Although our analysis to quantify the methane emissions can retain large relative confidence intervals, especially during periods of highly variable observations, an urban core site with precise instrumentation measuring multiple trace gas species can still be very informative, including potentially resolving diurnal methane emissions patterns previously not shown from afternoon-only studies.

Even the best performing methane emissions inventory, developed specifically for the NYCMA at higher resolution, underestimated the observed atmospheric methane during peak emission events in winter. These methane emission peaks were correlated with elevated CO emissions, which provides strong evidence that these unaccounted-for methane emissions are from the same stationary combustion source type as the CO. Clearly, there is a city-scale atmospheric impact of combustion emissions, but we do not know how widespread the source type is or if there are a few large source points or many small ones.

Examining the characteristics of the local-scale measurements removed from the analysis here may help answer these questions.

Future studies should also focus on approximating the stationary incomplete combustion sources based on relationships with temperature or other environmental factors and knowledge of urban stationary combustion systems (i.e., building boilers). Given the uncertainties previously discussed, we need additional sites with 24-hour year-round atmospheric measurements of methane and CO, collocated meteorological observations, and systematic evaluation of reanalysis and forecast products, especially at night, to improve the continuous quantification of methane and CO emissions and define their source apportionment. Discovering a mechanistic driver for the methane emissions variability related to incomplete combustion will allow for these emissions estimates to be improved and included in future inventories enabling stakeholders to properly target all potential methane emission sources and track and have confidence in the progress of greenhouse gas emission mitigation efforts.

# Data availability

Data that support the findings of this study are available as listed below:

ASRC Rooftop methane (CH<sub>4</sub>) observations and NYCMA observed and simulated  $\Delta$ CH<sub>4</sub>, with coincident carbon monoxide (CO) observations and  $\Delta$ CO [Dataset]. Dryad. https://doi.org/10.5061/dryad.ghx3ffc0g

Stockholm, New Jersey (SNJ) methane observations: https://data.nist.gov/od/id/mds2-3765 (Karion et al., 2025)

EDGAR v6.0 methane emissions: <a href="https://edgar.jrc.ec.europa.eu/dataset\_ghg60">https://edgar.jrc.ec.europa.eu/dataset\_ghg60</a>

665 EDGAR v8.0 methane emissions: https://edgar.jrc.ec.europa.eu/dataset\_ghg80

EPA GHGI v2016 methane emissions: https://www.epa.gov/ghgemissions/gridded-2012-methane-emissions (US EPA, 2016)

EPA GHGI v2023 and EPA GHGI v2023 EE methane emissions: https://zenodo.org/records/8367082 (McDuffie et al., 2023)

Pitt High-Resolution Inventory methane emissions: https://data.nist.gov/od/id/mds2-2915 (Pitt et al., 2024a)

EPA Cornwall CO observations: https://aqs.epa.gov/aqsweb/airdata/download files.html

EDGAR v8.1 CO emissions: https://edgar.jrc.ec.europa.eu/dataset\_ap81

STILT model: https://uataq.github.io/stilt/#/

HRRR ARL files: <a href="https://www.ready.noaa.gov/archives.php">https://www.ready.noaa.gov/archives.php</a>

#### **Author contributions**

670

LDS and RC designed the study. RC, AHD, and YZ provided calibrated ASRC methane and CO measurements. RTC supported ASRC measurements. LDS performed STILT simulations and primary analysis of observation-model comparison. RC and YZ assisted the analysis. LDS wrote the paper. All co-authors contributed to the preparation of the manuscript.

# **Competing interests**

Authors declare that they have no competing interests.

# Acknowledgements

This work was supported by the New York State Energy Research and Development Authority (NYSERDA), contract (#183867), by NOAA research grants (NA20OAR4310306; NA21OAR4310235), and by the Columbia University Department of Earth and Environmental Sciences. The authors thank John Mak, Lee Murray, Anna Karion, and Cody Floerchinger for instrumentation and measurement support, the Advanced Science Research Center (ASRC) for hosting the measurements at the ASRC Rooftop Observatory, and Bronte Dalton and Savannah Ferretti for helpful discussions. We also thank the STILT development team and the R Project community for analysis and plotting tools, especially the ggplot2, ggpattern, magick, anytime, lubridate, raster, and cowplot packages.

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
