# Peer review of "Missing wintertime methane emissions from New York City related to combustion"

_EGUsphere, 2025_

## Author Comment (AC2)

Response to Schiferl et al., ACPD, 2025

Reviewer 1:

The authors present multiple years of measured CH4 and CO concentrations from New York City, and combine the results with Lagrangian model output to infer emissions for both species. They compare the derived fluxes with inventory predictions and draw conclusions regarding patterns of variability and drivers of emissions.

The topic is well-suited to ACP and the findings are broad interest. I have listed comments below that I feel should be addressed prior to its publication.

We thank the reviewer for taking the time to provide comments that have greatly improved this manuscript. We have addressed each point below in blue with the text edits in bold.

Scientific comments
* * *
1.  A general question has to do with spatial representativeness. It seems likely that the inventories being examined have biases that vary in space, whereas the results from this particular rooftop are discussed as reflecting the NYCMA. How can we assess the representativeness of these findings? For example, does the inferred emission disparity for a given inventory vary with wind direction?

We thank the reviewer for highlighting this point.

The CUNY ASRC site has been used extensively in recent years as a site representative of high-density urban air. It has been the site of long-term studies (Schiferl et al., 2024), instrument characterization studies (Commane et al., 2023, Khare et al., 2022), COVID activity change studies (Cao et al., 2023, Tzortziou et al., 2022) and was the location of atmospheric chemistry focused intensive studies in 2022 and 2023 (e.g. Hass-Mitchell et al., 2024 for the NYC-Mets project). We have added the following text to the methods section (near line 93 in the original text) to highlight the previous studies that have been conducted at the CUNY ASRC site:

"The CUNY ASRC site has been used extensively in recent years as a site representative of high-density urban air around the New York City metropolitan area. It has been the site of long-term studies (Schiferl et al., 2024), instrument characterization studies (Commane et al., 2023, Khare et al., 2022), COVID activity change studies (Cao

et al., 2023, Tzortziou et al., 2022) and was the location of atmospheric chemistry focused intensive studies in 2022 and 2023 (e.g. Hass-Mitchell et al., 2024 for the NYC-Mets project)."

As seen in Figure S7, by aggregating multiple years of monthly data, the ASRC site is influenced by all wind directions including by the areas with the largest and most heterogeneous methane emissions in the Pitt inventory (Figure 4a). However, one limitation of aggregating at least 10 days or a month of observations to ensure representativeness is that a wind direction break-down of the inferred emissions is not possible. However, future work that will conduct a full geostatistical inverse analysis using additional tower observations should be able to assess the spatial distribution of the missing fluxes.

References:

Cao, C., Gentner, D. R., Commane, R., Toledo-Crow, R., Schiferl, L. D., and Mak, J. E.: Policy-Related Gains in Urban Air Quality May Be Offset by Increased Emissions in a Warming Climate, Environ. Sci. Technol., 57, 9683–9692, https://doi.org/10.1021/acs.est.2c05904, 2023.

Commane, R., Hallward-Driemeier, A., and Murray, L. T.: Intercomparison of commercial analyzers for atmospheric ethane and methane observations, Atmos. Meas. Tech., 16, 1431–1441, https://doi.org/10.5194/amt-16-1431-2023, 2023.

Hass-Mitchell, T., Joo, T., Rogers, M., Nault, B. A., Soong, C., Tran, M., Seo, M., Machesky, J. E., Canagaratna, M., Roscioli, J., Claflin, M. S., Lerner, B. M., Blomdahl, D. C., Misztal, P. K., Ng, N. L., Dillner, A. M., Bahreini, R., Russell, A., Krechmer, J. E., Lambe, A., and Gentner, D. R.: Increasing Contributions of Temperature-Dependent Oxygenated Organic Aerosol to Summertime Particulate Matter in New York City, ACS EST Air, 1, 113–128, https://doi.org/10.1021/acsestair.3c00037, 2024.

Khare, P., Krechmer, J. E., Machesky, J. E., Hass-Mitchell, T., Cao, C., Wang, J., Majluf, F., Lopez-Hilfiker, F., Malek, S., Wang, W., Seltzer, K., Pye, H. O. T., Commane, R., McDonald, B. C., Toledo-Crow, R., Mak, J. E., and Gentner, D. R.: Ammonium adduct chemical ionization to investigate anthropogenic oxygenated gas-phase organic compounds in urban air, Atmos. Chem. Phys., 22, 14377–14399, https://doi.org/10.5194/acp-22-14377-2022, 2022.

Schiferl, L. D., Cao, C., Dalton, B., Hallward-Driemeier, A., Toledo-Crow, R., and Commane, R.: Multi-year observations of variable incomplete combustion in the New York megacity, Atmos. Chem. Phys., 24, 10129–10142, https://doi.org/10.5194/acp-24-10129-2024, 2024.

Tzortziou, M., Kwong, C. F., Goldberg, D., Schiferl, L., Commane, R., Abuhassan, N., Szykman, J. J., and Valin, L. C.: Declines and peaks in NO2 pollution during the multiple waves of the COVID-19 pandemic in the New York metropolitan area, Atmos. Chem. Phys., 22, 2399–2417, https://doi.org/10.5194/acp-22-2399-2022, 2022.

2. Abstract: "The estimates of methane emissions correlated with those of carbon monoxide (CO) emissions, determined from coincident measurements, suggesting a common city-scale incomplete combustion source for both methane and CO." And line 554, "The afternoon observation-informed methane and CO emissions rates for the NYCMA were well correlated over our study period (Fig. 6c, $R^2$ = 0.59). Unlike the observed $\Delta CH_4:\Delta CO$ comparison in Sec. 3.1, this relationship between methane and CO emissions accounted for variability in atmospheric transport."

   But to what degree could this correlation reflect footprint effects? E.g., if there is more developed infrastructure in upwind direction A than B, we would expect both CO and CH4 to be enhanced when winds are from A and not when winds are from B. This could drive a correlation that doesn't necessarily reflect a mechanistic source connection.

The reviewer raises a good point about the impact of wind direction and the mechanism of the methane sources.

We used monthly time averaging to calculate the emissions, which smoothed out any spatial heterogeneity in emissions within the monthly footprint. Figure S7 shows the common sampling area for each monthly footprint (surface influence functions) for each year of observations. While it is likely that there is spatial heterogeneity in methane and CO emissions driven by the high and low urban density, our analysis cannot resolve these higher spatial resolution processes. We also note that Fig 6c shows the domain-averaged monthly emissions independently calculated for methane and CO.

While the correlation of CO and methane emissions does not necessarily indicate a mechanistic source connection, the correlation of methane and CO emissions with seasonal changes (maximum in January/February to minimum in May) indicates a co-located source of methane and CO with a similar temporal profile to that of the heating season and natural gas consumption/combustion in the region.

3. Can you test the reasoning here using your model simulations? Your inventories segregate emissions by sector. So one can track the modeled CH4 and CO concentrations at the receptor that arise from different source sectors. If the reasoning above is sound, then we would not see a correlation between the

modeled CO concentrations and the modeled non-combustion CH4 concentrations.

We disagree with the reviewer on this point. Atmospheric enhancements are fundamentally different from emissions. Changes in the atmospheric transport and vertical mixing (i.e. mixed layer height and wind direction) drive most of the variability in both the observed and simulated enhancements.

The spatial distribution of the CO and methane inventories are broadly similar in the New York City metropolitan area as both inventories distribute their emissions according to the population density. So, other than a few point sources of methane, we would generally expect a correlation between the simulated non-combustion methane and CO enhancements. Excluding the 5-10% of methane emissions due to stationary combustion (Table 1) from the inventory would not remove the correlation in the simulated enhancements of CO and methane.

To address both point 2 and 3, we have added the following text:

"The afternoon observation-informed methane and CO emissions rates for the NYCMA were well correlated over our study period (Fig. 6c, $R^2 = 0.59$). These emission rates account for variability in atmospheric transport and vertical mixing over monthly time scales, as shown in the spatial distribution of the footprints (Figure S7). While the correlation of CO and methane emissions does not necessarily indicate a co-emitted source, the seasonal changes (maximum in January/February to minimum in May) indicates a co-located source of methane and CO with a similar temporal profile to that of natural gas consumption/combustion in the region. We do not know the $CH_4:CO$ emission ratio, nor the modified combustion efficiency of individual incomplete combustion sources (i.e. boilers and other appliances) within our study domain. However, we can expect the $CH_4:CO$ ratio to be variable with each appliance configuration, and so it may change across time and space.

We likely observed two competing thermogenic methane source sectors at the rooftop: (1) inefficient consumption of natural gas during peak heating season (January-February), which is correlated with extreme cold events, and (2) intermittent emissions of natural gas during the appliance duty cycle (also known as "slip") (Lindburg et al., 2025). During the winter-to-spring transition, when outdoor temperatures vary around 55°F, the threshold below which all buildings are required to be heated by NYC laws (Chapter 2: Housing Maintenance Code, 2025), boilers will repeatedly cycle. This evidence suggests a common source of methane and CO emissions, which may be related to stationary incomplete combustion. Further study is required to isolate and quantify these processes in more detail."

4. Lines 449-456: I am not totally sold by the reasoning in this section, in two respects. First, "Accounting for the varying atmospheric transport and mixing throughout the study period, which drives nearly all variability in the simulated ΔCH4, we found that meteorology only explained 30%–43% of the variability in observed ΔCH4, depending on inventory comparison, based on the calculated R2 between the observed ΔCH4 and simulated ΔCH4". To me, "meteorology" as used in this context implies dilution/ventilation/mixing effects. But there is also

the fact that changing wind directions also change the portion of the city that is being sampled. So the wording should be more precise here.

We thank the reviewer for highlighting that lack of clarity. We address the changing wind directions above. In this case we meant meteorology to include both horizontal atmospheric transport and vertical mixing (including dilution and ventilation). The wind direction changes over either the 10 days or monthly time periods explored here are a key component of this analysis. To remove any confusion, we have modified the text to say atmospheric transport and vertical mixing instead of meteorology in this section.

Line 449: "Accounting for the varying atmospheric transport and **vertical** mixing"

Line 450: "we found that **atmospheric transport and vertical mixing** only explained"

Line 454: "the impact of **atmospheric transport and vertical mixing** on observed ΔCH4"

Line 456: "was due to atmospheric transport **and vertical mixing**".

5. Second, based on the argument that the variability in simulated concentrations is entirely due to transport effects, and the fact that the model and observations correlate to ~ R2 0.3-0.43, the authors infer that meteorology explains 30-43% of the variability in the CH4 observations. Since the models correlate more strongly with CO, the authors then posit that "Therefore, methane emissions vary more than CO emissions on a 10-day time scale." But the footprint issue is relevant here as well. For example, if the spatial distribution of CO emissions in the inventories were accurate, and the spatial distribution for CH4 was not, then the model-measurement correlation for CO will be higher than for CH4, independent of any potential temporal variability.

We thank the reviewer for highlighting this point and we both agree and disagree with the reviewer here. In general, both the CO and methane inventories are distributed across population density so both are just as likely to be spatially accurate. The various methane inventories have slightly different spatial distributions of methane emissions across the domain (see Fig S4 and S5) and yet the simulated enhancements from each inventory have similar correlation coefficients with the observed enhancements (0.3 - 0.43), which would imply that the footprints (especially the vertical mixing) are driving much of the correlation, rather than the spatial distribution of the inventories themselves.

Aside from stationary combustion, most of the CO emissions do not change much on a 10 day or seasonal time scale. These stable CO emission sectors (traffic, power generation, manufacturing) would not be expected to contribute much to the methane emissions totals for the NYCMA.

We have edited the text in blue to clarify this point:

"We found that the impact of atmospheric transport and vertical mixing on observed $\Delta CH4$ in this study was considerably less than was found by Schiferl et al. (2024) for CO using the same metric and largely same methods, where ~60% of the variability in observed $\Delta CO$ was due to atmospheric transport and vertical mixing. The weaker correlation for methane than CO implies that the methane emissions may change more across the seasons when calculated on a 10-day time scale. This result is consistent with relatively unchanging seasonal magnitudes of CO emissions from traffic, power generation, and manufacturing, all of which are sources that are not expected to contribute much to the methane emissions totals in the NYCMA."

6. Lines 110-120: separation of local-scale vs. city-scale influences. Hours with SD(CO) < 200 ppb will still be influenced by (albeit slightly smaller) local sources. It seems that the threshold choice here is fairly arbitrary. I recommend including a sensitivity analysis varying this threshold to demonstrate (hopefully) that the specific choice does not have a major effect on the findings.

We thank the reviewer for highlighting this point. The choice of a CO standard deviation of < 200 ppb was based on a detailed analysis of that threshold undertaken as part of Schiferl et al., 2024. 200 ppb gave similar results as the two-tower approach used in that study. As also written below in response to Reviewer 2, we have included the text:

Original Line 113: "In the 1-Hz data, all examples of highly variable methane plumes (ie. near field sources) were strongly correlated with highly variable CO ($R^2 > 0.99$)."

Original Line 115: "The threshold of 200 ppb for the CO standard deviation was chosen from a sensitivity analysis to replicate the results of the two-tower approach detailed in Schiferl et al., 2024."

7. Line 281: "Interaction between the surface flux and atmospheric mole fraction (the surface influence) happens when particles are present within the lower half of the mixing layer."

Same comment, this choice is also somewhat arbitrary. I have no objection to the particular choice, and I think it is a standard value used in STILT work. But can we demonstrate what the impact of this assumption is via the sensitivity analyses?

We thank the reviewer for highlighting this point too. The choice of the lower half of the mixing height is indeed arbitrary but it is the default in STILT. We evaluated parameters in the HRRR-STILT model that would simulate even greater changes than expected from changing this fraction.

As part of other studies, we evaluated four different meteorological products and undertook two different sensitivity analyses to evaluate the default settings used in STILT by calculating the simulated mixing ratio enhancement due to each configuration. In Schiferl et al., 2024, we found similar variability of simulated CO enhancement when using HRRR (3 km) and NAMS (12 km) meteorological products but spatially coarser meteorological products like GFS (~28 km) and GDAS (2.5 degrees) underpredicted the CO enhancement by ~30 ppbv. We examined the sensitivity of the simulated CO enhancement to many configuration variables in STILT-HRRR and found that only the minimum Mixing Layer Height (MLH) produced a quantifiable change (> 1 ppbv) in the simulated CO mixing ratio; a 20 ppbv increase in simulated CO enhancement was observed when reducing the minimum MLH from 250 m to 150 m (Figure S7 in Schiferl et al., 2024). Recently we evaluated four possible parameterizations of the MLH in STILT, (a) embedded HRRR, (b) temperature profile method, (c) turbulence kinetic energy method, and (d) default Richardson number method. All configurations simulated methane enhancements that differed by less than 1 ppbv in the afternoon, increasing to 5 ppbv at night.

We have added the following text after line 300:

While testing the configuration, Schiferl et al. (2024) found that the model configuration for vertical mixing and choice of meteorological product had little effect on the results at this site. They found that only the choice of the minimum Mixing Layer Height (MLH) produced a quantifiable change (> 1 ppbv) in the simulated CO mixing ratio; a 20 ppbv increase in simulated CO enhancement was observed when reducing the minimum MLH from 250 m to 150 m (Figure S7 in Schiferl et al., 2024). We evaluated four possible parameterization of the MLH in STILT and all configurations simulated methane enhancements that differed by less than 1 ppbv in the afternoon, increasing to a maximum of 5 ppbv at night. We also tested the impact of both the STILT minimum mixing height (150m v. 250m) and meteorological products (HRRR v. NAMS, North American Mesoscale Forecast System at 12 km horizontal resolution) on our monthly observation-informed emissions estimates (see Sec. 2.5) for 2023 and 2024 and discuss those sensitivity results in Sec. 3.3.

8. Eq. 3. I agree that we expect the relative concentration disparity to correspond to the relative emission disparity. But you also have a nice opportunity to demonstrate that to the reader rather than just asserting it… i.e. if you scale emissions by X do your HRRR-STILT concentrations at the receptor increase by the same factor X?

The reviewer makes an interesting point here. As part of the development of our diurnal analysis framework, we tested this assumption. We applied the scaling factor to the simulated methane enhancements for Jan 1 - March 14, 2020 for the mean diel cycle. Depending on the inventory used, the scaling factor varied from 1.5-2 at night to 2.8-4.5 during the afternoon. The scaled simulated enhancement matched the observed methane enhancement well for both weekday (solid filled) and weekend (open symbols) across the diel cycle. Below we show the diel cycle of the observed (black) and simulated (blue, green, red) for spring 2020 (before COVID) on the left, and on the right, the scaling factor for the observed/simulated. We selected the median inventory (EDGAR 6.0) and the bottom plot shows how the methane enhancement calculated from the scaled up EDGAR emissions matches the observations.

[Figure]

9. Line 483: "Since there was no diurnal variability in the inventory methane emissions, the diurnal variability in the simulated ΔCH4 was entirely due to changes in the surface influence footprint (i.e., transport meteorology) throughout the day. The differences in the variability between the observed ΔCH4 and simulated ΔCH4 were therefore due to changes in the methane emissions which were not included in the inventory"
And Line 595: "The consistent difference between the afternoon and 24-hour emission rates suggests a diurnal cycle in emissions, which is well-known for CO emissions (traffic, human activity), but had not been, to our knowledge, previously inferred for urban methane emissions."
Couldn't a diurnally-dependent bias in the model met fields also be a plausible way to explain this? It seems likely that this occurs at least to some extent.

We thank the reviewer for highlighting this point, which was also highlighted by Reviewer 2. Yes, the calculation of the observed/simulated ratio assumes there is limited bias in the HRRR-STILT calculated footprints over the diel cycle. Sensitivity tests of the HRRR-STILT simulations found that simulated methane could change by up to 1ppb during the afternoon to 5 ppb overnight depending on the configuration of the model. However, the diel changes in the methane enhancements are of the order 80-200 ppb at night (Figure 5a) so the bias is at most 6%. We would expect if uncertainty in the transport model is dominating the diel cycle that the CO and methane fluxes would be influenced in the same way. However, the calculated CO emissions showed a traffic dependent diel cycle (outside of the COVID lockdown), which suggests that the uncertainty in the transport model is not the limiting factor.

We added the blue text after line 300 to highlight this point about the systematic bias:

"While testing the configuration, Schiferl et al. (2024) found that the model configuration for vertical mixing and choice of meteorological product had little effect on the results at this site. They found that only the choice of the minimum Mixing Layer Height (MLH) produced a quantifiable change (> 1 ppbv) in the simulated CO mixing ratio; a 20 ppbv increase in simulated CO enhancement was observed when reducing the minimum MLH from 250 m to 150 m (Figure S7 in Schiferl et al., 2024). We evaluated four possible parameterization of the MLH in STILT and all configurations simulated methane enhancements that differed by less than 1 ppbv in the afternoon, increasing to a maximum of 5 ppbv at night. We also tested the impact of both the STILT minimum mixing height (150m v. 250m) and meteorological products (HRRR v. NAMS, North American Mesoscale Forecast System at 12 km horizontal resolution) on our monthly observation-informed emissions estimates (see Sec. 2.5) for 2023 and 2024 and discuss those sensitivity results in Sec. 3.3."

We also added the blue text around Figure 5 to highlight that there may be limited bias from the transport model.

"Since there was no diurnal variability in the inventory methane emissions, the diurnal variability in the simulated $\Delta CH_4$ was entirely due to changes in the surface influence footprint (i.e., atmospheric transport and vertical mixing) throughout the day. Sensitivity studies of the simulated footprints found that simulated methane could change by up to 1ppb during the afternoon to 5 ppb overnight depending on the configuration of the model. However, the diel changes in the methane enhancements were of the order 80-200 ppb at night (Figure 5a) so we estimated the model bias was at most 6%. The differences in the variability between the observed $\Delta CH_4$ and simulated $\Delta CH_4$ were therefore attributed to changes in the methane emissions which were not included in the inventory. By normalizing the observed $\Delta CH_4$ by the simulated $\Delta CH_4$, we minimized the impact of meteorology, thereby isolating only the changes in methane emissions."

Technical comments
* * *
10. Line 33: grammar, "highlighting the importance of accurately quantify"

**We have clarified the text:**

Recently, methane emissions from oil and gas infrastructure (i.e., rural production facilities, pipeline leaks in cities) have received particular attention as mitigation targets (Alvarez et al., 2018; Ocko et al., 2021), highlighting the importance of accurately quantifying baseline methane emissions in order to track the effectiveness of mitigation efforts.

11. Line 38: awkward, "identified atmospheric methane greater than expected"

**We have clarified the text:** "Previous studies  have identified atmospheric methane emissions that were greater than expected from inventories for cities across the world …"

12. Line 58: suggest "reduced THE ESTIMATED methane emissions

**We have clarified the text:** "The most recent US national gridded inventory (EPA GHGI v2023, Maasakkers et al., 2023) reduced the estimated methane emissions relative to the previous version (EPA GHGI v2016, Maasakkers et al., 2016) for the NYCMA, thus worsening the underestimate."

13. Lines 135-140: was this a simple boxcar average or something else?

**We don't exactly know what the reviewer means by a boxcar average (we suspect the answer is yes). We calculated a 10-day centered mean value (or 5%ile) for each hour.**

14. Lines 167-169: I find the wording here confusing. Do you just mean you don't average the same time period twice when computing averages? Please clarify wording

**We thank the reviewer for catching this. We have clarified the text.**

"To avoid biasing the corresponding distributions of mean observed $\Delta CH_4$ and $\Delta CO$, we only used a given background site type (urban core or remote) in the distribution when  both methane and CO data were available."

15. 487: "When tested for …" this sentence is not very clear as written, can it be clarified?

We have clarified the text.

"Using this method, Schiferl et al. (2024) found that most of the CO emissions changes occurred in areas located within 2 hours atmospheric transport of the ASRC site and we expect a similar atmospheric transport time for methane. This normalization method produced observation-informed changes in methane emissions for multi-week periods before and during the COVID-19 shutdown (Fig. 5b)."
* * *
Reviewer 2:

In "Missing wintertime methane emissions from New York City related to combustion," the authors analyze rooftop measurements of methane and carbon monoxide to investigate the variability of methane emissions and extend the time frame of a previous carbon monoxide study. The work presented is thorough and the paper is well structured, well written, and provides sufficient background and context for how the work fits into the broader field. Urban methane has received increased attention in recent years and many important scientific questions remain around temporal variability and source attribution. Accordingly, I find the presented work to be timely and of likely interest to the ACP reader.

We thank the reviewer for taking the time to provide comments that have greatly improved this manuscript. We have addressed each point below in blue with the text edits in bold.

1. I believe clarifications around some conclusions are required prior to publication:

   1. Influence of transport/spatial sensitivity

The authors use the STILT model run by HRRR met fields to calculate the surface influence (footprint) of their observations at the rooftop site. This method is well established in the literature. They also provide the footprints (both afternoon and 24-hours) in the supplement at the monthly scale. At the monthly scale (Figure S7), differences can be seen between the spatial sensitivity of the site between different months and different years. I would expect further differences in spatial sensitivities to emerge at shorter time scales, such as the 10-day and 2-hourly intervals used in this work (e.g., Figure 5). Are the conclusions drawn from these shorter time intervals robust to differences in spatial sensitivity from one interval to the next?

We thank the reviewer for highlighting this point.

The 10-day methane and CO enhancements were surprisingly strongly correlated (Fig 2a), but we had to account for atmospheric transport (horizontal and vertical) before we could ascribe this correlation to emissions. We calculated the monthly emissions of $CH_4$ and CO by accounting for atmospheric transport (using the HRRR-STILT footprints) and found that the $CH_4$ and CO observation-informed emissions were also correlated. While the 10-day and monthly time averages sample different areas, the consistency is likely due to the urban core with the largest methane and CO emissions being uniformly sampled for all times, with different areas of lower methane and CO emissions being sampled by the outer areas of the footprints each month.

We only used 2 hourly intervals to examine diurnal cycles of either an entire month of data or two-week time periods during COVID (ie. all hours at 0:00, 1:00 for an entire month were averaged together). We have updated the text to clarify that point.

Line 469: "mean 2-hour" replaced by "mean diel cycle of"

2. In addition, for the diurnal time series shown in Figure 5b the ratio of the observed/simulated is assumed to remove the influence of transport. But does that not require that the representation of the transport be completely accurate? Could there still be transport related signals in the ratio when HRRR-STILT does not capture the transport sufficiently? Particularly at night.

We thank the reviewer for highlighting this point. Yes, the calculation of the observed/simulated ratio assumes there is limited bias in the HRRR-STILT calculated footprints. Sensitivity tests of the HRRR-STILT simulations found that simulated methane could change by up to 1ppb during the afternoon to 5 ppb overnight depending on the configuration of the model. However, the diel changes in the methane enhancements are of the order 80-200 ppb at night (Figure 5a) so we estimate that the bias is at most 6%.

We added the blue text after line 300 to highlight this point about the systematic bias:

"While testing the configuration, Schiferl et al. (2024) found that the model configuration for vertical mixing and choice of meteorological product had little effect on the results at this site. They found that only the choice of the minimum Mixing Layer Height (MLH) produced a quantifiable change (> 1 ppbv) in the simulated CO mixing ratio; a 20 ppbv increase in simulated CO enhancement was observed when reducing the minimum MLH from 250 m to 150 m (Figure S7 in Schiferl et al., 2024). We evaluated four possible parameterization of the MLH in STILT. All configurations simulated methane enhancements that differed by less than 1 ppbv in the afternoon, increasing to a maximum of 5 ppbv at night. We also tested the impact of both the STILT minimum mixing height (150m v. 250m) and meteorological products (HRRR v. NAMS, North American

Mesoscale Forecast System at 12 km horizontal resolution) on our monthly observation-informed emissions estimates (see Sec. 2.5) for 2023 and 2024 and discuss those sensitivity results in Sec. 3.3."

We also added the following blue text around Figure 5 to highlight that there may be limited bias from the transport model.

"Since there was no diurnal variability in the inventory methane emissions, the diurnal variability in the simulated $\Delta CH_4$ was entirely due to changes in the surface influence footprint (i.e., atmospheric transport and vertical mixing) throughout the day. Sensitivity studies of the simulated footprints found that simulated methane could change by up to 1ppb during the afternoon to 5 ppb overnight depending on the configuration of the model. However, the diel changes in the methane enhancements were of the order 80-200 ppb at night (Figure 5a) so we estimated the model bias was at most 6%. The differences in the variability between the observed $\Delta CH_4$ and simulated $\Delta CH_4$ were therefore attributed to changes in the methane emissions which were not included in the inventory. By normalizing the observed $\Delta CH_4$ by the simulated $\Delta CH_4$, we minimized the impact of meteorology, thereby isolating only the changes in methane emissions."

3. CO as a combustion tracer

In section 2.6, CO is described as a combustion tracer. However, this section goes on to discuss how the source of CO and methane are not co-located for typical city sources. Based on this section, I am not quite clear how CO can act as a methane combustion tracer at the city-scale.

We thank the reviewer for highlighting this oversight. Typical methane sources (e.g. natural gas leaks or emissions from wastewater treatment plants) are not often known to co-emit CO with methane. However, some studies have suggested a link between methane and CO sources in urban areas (e.g. Helfter et al., 2016, Pawlak et al., 2016). Leaks of methane from natural gas pipeline infrastructure (pre or post-meter) should not be correlated with CO. However, incomplete combustion of post-meter natural gas (which produces CO) can lead to large methane emissions, especially from residential and commercial boilers (Lindberg et al., 2025). We also observed strongly correlated methane and CO dry mole fractions in the 1-Hz data observed at ASRC ($R^2 > 0.99$), suggesting the near field local sources were related to combustion.

We have clarified and added the following text to Section 2.6 (near Line 328):

"Aside from wildfires and wood burning, methane and CO sources are not usually linked in emissions inventories. Generally, CO emission sources are not co-located with large urban methane emission sources such as landfills, wastewater treatment, and natural gas distribution, as the methane from these sources is not actively being combusted. However, some studies have suggested a link between methane and CO sources in urban areas (e.g. Helfter et al., 2016, Pawlak et al., 2016). Leaks of methane from natural gas pipeline infrastructure (pre or post-meter) should not be correlated with CO. However, incomplete combustion of post-meter natural gas (which produces CO) can also lead to large methane emissions, especially from residential and commercial boilers (Lindberg et al., 2025). In the 1-Hz methane and CO observed at ASRC, all examples of highly variable methane

plumes were strongly correlated with highly variable CO ($R^2 > 0.99$), suggesting that the observed near field sources were related to combustion."

4. When discussing the correlation between observed enhancements of methane CO (Lines 376-375), the authors note that "a large portion of the correlation is likely from the variability of atmospheric transport but could also indicate simultaneous emission sources of methane and CO." I agree. Later, the correlation of methane and CO emissions rates (~Line 555) is used as possible evidence for a common city-scale incomplete combustion source. The reasoning is based on the use of HRRR-STILT to calculate both emission rates. Similar to my point (1), isn't there the possibility that influences from transport could remain? Differences in the accuracies of spatial allocations in urban CO and methane inventories could also influence the emission rates due to their use in calculating the simulated enhancements. Perhaps there is a way to assess the sensitivity of the emission correlations to these influences.

We thank the reviewer for mentioning this point. In general, both the CO and methane inventories are distributed across population density so both are just as likely to be accurately distributed spatially, if not correct in magnitude. There is a possibility that the influence from transport could remain but any vertical mixing bias should affect both gases equally. Any spatial bias in the transport model should be averaged out through our use of monthly aggregated footprints for our emissions estimates (Fig S7).

The inventories are used as a spatial prior when they are convolved with the footprints but the emissions are calculated over the whole domain using the observations for each month. We calculated very different scaling factors for each of the various methane inventories, which have different spatial distributions of methane emissions, and did not find any significant difference in the calculated emissions resulting from each simulation.

5. Uncertainty of 24-hour data

As the authors state, "Meteorological products (i.e., 3-dimensional wind fields) used to drive the atmospheric transport model are more uncertain at night, for mixing heights especially, ..." (Lines 518-519). It is not clear to me if this larger uncertainty in transport at night is incorporated into the confidence intervals for the 24-hour emission estimates or diurnal analysis.

It is difficult to quantify the bias that each met product might have for night-time simulated data so we do not consider increased transport uncertainty at night in the total uncertainty calculation. Instead, the bootstrap method subsamples all data within 24 hours so that the uncertainty includes the variability of the background uncertainty, the variability of the observed methane mole fractions within each period, and the ensemble estimate from the Pitt High-Resolution Inventory.

We have added text that acknowledges that point.

Line 518: "As it was difficult to quantify any night-time bias in the meteorological product, we did not include this factor in the uncertainty calculation."

6.  It is stated that for 24-hours "Longer aggregation time resulted in much narrower confidence intervals…." (Lines 323-326) and "The combination of many hours to produce the 24-hour emissions estimates resulted in much smaller confidence intervals compared to the afternoon-only emission estimates, further supporting the possibility of diurnal cycle for urban methane emissions" (Lines 600-602). More data would likely lead to higher confidence in the estimates, but does this uncertainty approach consider the increased transport uncertainty in the non-afternoon hours?

We do not include the increased transport uncertainty in the non-afternoon hours. Repeating point 5 above:

It is difficult to quantify the bias that each met product might have for night-time simulated data so we do not consider increased transport uncertainty at night in the total uncertainty calculation. Instead, the bootstrap method subsamples all data within 24 hours so that the uncertainty includes the variability of the background uncertainty, the variability of the observed methane mole fractions within each period, and the ensemble estimate from the Pitt High-Resolution Inventory

We have clarified the text around line 600 in the original text to:

"The combination of many hours to produce the monthly 24-hour emissions estimates resulted in narrower confidence intervals compared to the afternoon-only emission estimates, further supporting the possibility of diurnal cycle for urban methane emissions."

Minor comments:

7. Lines 113-120: Methane categorization as local- or city-scale. Variability in CO at an hourly timescale is used to remove near-field signals? Is this a typo, should it this methane instead of CO? In the near-field methane and CO might not be co-emitted.

We thank the reviewer for highlighting this point. The use of CO is not a typo but we have added more text to clarify the choice of CO.

In the 1Hz data collected in this study, all examples of highly variable methane plumes (ie. near field local sources) were correlated with highly variable CO. The choice of a CO standard deviation of < 200 ppb was based on a detailed analysis of that threshold undertaken as a complement to Schiferl et al., 2024. 200 ppb gave similar results to that of the two-tower approach in Schiferl et al., 2024. As written above in response to Reviewer 1, we have included the text:

Original Line 113: "In the 1-Hz data, all examples of highly variable methane plumes (ie. near field sources) were strongly correlated with highly variable CO ($R^2 > 0.99$)."

Original Line 115: "The threshold of 200 ppb for the CO standard deviation was chosen from a sensitivity analysis to replicate the results of the two-tower approach detailed in Schiferl et al., 2024."

8. Lines 249-255: "… we chose to leave emissions constant throughout the day …" It is not clear to what this statement applies. Is what part of the analysis are emissions assumed to be constant throughout the day? I assume it to calculated the simulated enhancements, but it is not clear from the text.

The reviewer is correct. We did not apply the Crippa diurnal correction to any of the inventories. We have clarified the text to say:

"While Crippa et al. (2020) suggested methods to implement diurnal variability in EDGAR using nationwide sector-specific scale factors, we did not apply a diel correction to the emissions of any inventory. Emissions for all inventories were constant throughout the day."

9. Line 556: "CH4:CO emission ratio" of combustion appliances I assume?

Yes. We have clarified the text to say:

"We do not know the $CH_4$:CO emission ratio, nor the modified combustion efficiency of individual incomplete combustion sources (i.e. boilers and other appliances) within our study domain."

**\*\*\*\*\*\*\*\*\*\***

Comment 1 by Thomas Karl:

The paper by Schiferl et al. identifies and quantifies additional anthropogenic sources of methane in New York, presenting evidence that these sources likely originate from incomplete combustion. This is an important finding, as it emphasises the need to consider emissions other than leakage alone. Various direct long-term eddy covariance observations of methane in Europe have reached a similar conclusion, showing that methane emissions follow a negative temperature dependence, which can not be explained by pure gas leaks. See, for example,

Helfter et al. (doi: 10.5194/acp-16-10543-2016).

Stichaner et al., (doi: 10.1016/j.atmosenv.2024.120743).

Pawlak et al.,(doi: 10.5194/acp-16-8281-2016).

Stichaner et al. showed that the temperature dependence of urban methane fluxes corresponds well with gas consumption data (ie. colder temperatures = more methane consumption), suggesting that pre-flush operation can lead to the release of unburned methane. Pure methane leaks would not be expected to exhibit a significant temperature dependence.

Based on the findings presented by Schiferl et al., there seems to be growing evidence that this phenomenon is not limited to a single city or location, but is instead generally linked to the operational conditions of gas furnaces and appliances. If generalized, a significant fraction of methane emissions may therefore be released at rooftop level above street canyons through chimneys and smoke stacks.

We thank Dr. Karl for this helpful comment. We appreciate his knowledge and wider context. There is indeed a growing body of evidence that incomplete combustion may be a significant driver of urban scale methane emissions across continents. We as a community may need to reconsider our sampling strategies (street sampling vs rooftops) to ensure we capture these emissions. We have included these citations and apologize for missing them the first time.